# A Systematic Review of the Current Status of Magnetic Resonance–Ultrasound Images Fusion Software Platforms for Transperineal Prostate Biopsies

**DOI:** 10.3390/cancers15133329

**Published:** 2023-06-24

**Authors:** Nahuel Paesano, Violeta Catalá, Larisa Tcholakian, Enric Trilla, Juan Morote

**Affiliations:** 1Clínica Creu Blanca, 08034 Barcelona, Spain; violetacatala@uroima.com (V.C.); larisa.tcholakian@creublanca.es (L.T.); 2Department of Urology, Vall d’Hebron Hospital, 08035 Barcelona, Spain; enrique.trilla@vallhebron.cat (E.T.); juan.morote@vallhebron.cat (J.M.); 3Department of Surgery, Universitat Autònoma de Barcelona, 08193 Bellaterra, Spain

**Keywords:** systematic review, robot-assisted prostate biopsy, MR-target transperineal biopsy, software-assisted MRI–TRUS fusion platforms, prostate cancer

## Abstract

**Simple Summary:**

With the consolidation of multiparametric resonance of the prostate as an effective method in the diagnosis of clinically significant prostate cancer (csPCa), robot-assisted devices have been developed in recent years to use in targeted prostate biopsy. However, their potential advantages over standard biopsy remain unclear. The 2019 European Association of Urology (EAU) prostate cancer (PCa) guidelines recommend transperineal biopsy as the first option over transrectal biopsies. From this systematic review, we confirm that almost with all developed devices, a series of transperineal biopsies have been reported. Those using rigid fusion systems have reported better detection rates of csPCa.

**Abstract:**

Given this new context, our objective is to recognize the suitability of the currently available software for image fusion and the reported series using the transperineal route, as well as to generate new evidence on the complementarity of the directed and systematic biopsies, which has been established through the transrectal approach. Evidence acquisition: This systematic review, registered in Prospero (CRD42022375619), began with a bibliographic search that was carried out in PubMed, Cochrane, and Google Scholar databases. The Preferred Reporting Items for Systematic Reviews and Meta-analyses (PRISMA) criteria and the studied eligibility based on the Participants, Intervention, Comparator, and Outcomes (PICO) strategy were followed. Warp analysis of selected studies was performed using the Quality Assessment of Diagnostic Accuracy Studies (QUADAS-2) tool. In addition, a Google search of all currently available fusion platforms was performed. Our Google search found 11 different commercially available robots to perform transperineal image fusion biopsies, of which 10 devices have published articles supporting their diagnostic effectiveness in transperineal prostate biopsies. Results: A total of 30 articles were selected and the characteristics and results of the biopsies of 11,313 patients were analyzed. The pooled mean age was 66.5 years (63–69). The mean pooled PSA level was 7.8 ng/mL (5.7–10.8). The mean pooled prostate volume was 45.4 cc. (34–56). The mean pooled PSA density was 0.17 (0.12–0.27). The overall cancer detection rate for all prostate cancers was 61.4%, while for csPCa it was 47.8%. PCa detection rate was more effective than that demonstrated in the systematic transrectal biopsy. However, the detection of csPCa in the systematic biopsy was only 9.5% in the reported series. To standardize our review, we grouped prostate cancer screening results according to the population studied and the software used. When the same populations were compared between elastic and rigid software, we found that rigid biopsies had a higher csPCa detection rate than biopsies with elastic fusion systems. Conclusion: Platforms performing prostate biopsy using transperineal image fusion have better detection rates of csPCa than systematic transrectal biopsies. Rigid fusion systems have a better csPCa detection rate than elastic ones. We found no diagnostic differences between the different types of robotic systems currently available. The complementarity of systematic biopsy has also been demonstrated in transperineal imaging fusion biopsies.

## 1. Introduction

Prostate cancer (PCa) is currently the most common malignancy and the third leading cause of cancer mortality in men in the United States (US) and Europe (EU) [1]. It represents approximately 7.5% of the new cancer cases diagnosed in the EU [2]. Approximately 1.4 million men worldwide were diagnosed with PCa in 2020, representing 15% of all cancers in men [3]. The suspicion of PCa is based on the serum measurement of prostate-specific antigen (PSA) level and the abnormalities detected via the digital rectal examination (DRE), while prostate biopsy is conducted to confirm the diagnosis [4]. Prostate cancer screening continues to be controversial due to the adverse effects of excessive unnecessary prostate biopsies and the over-detection of insignificant tumors (iPCa), estimated to be between 40 and 50% of cases, which has often resulted in over-treatment [5].

The clinically significant PCa (csPCa) in prostate biopsies can be defined in several ways. The most widespread was established from the Gleason pattern 3 + 4 or higher, which is equivalent to the current groups 2 to 5 of the International Society of Urologic Pathology (ISUP-GG) [6]. In recent years, complementary tools have been suggested to avoid unnecessary biopsies and over-detection of iPCa, such as new molecular markers, predictive models, and classic PSA density. However, the greatest contribution has been made by the prostate magnetic resonance imaging (MRI) and the possibility derived from the biopsies of suspicious areas [7]. The development of multiparametric MRI techniques (mpMRI) and the overall improvement in their interpretation through the Prostate Imaging-Report and Data System (PI-RADS) have been essential to identify csPCa suspicious areas and their semi-quantitative risk [8,9]. These innovations and the evolution of the PI-RADS for the categorization of prostate lesions classified from very low to high suspicion of csPCa, have prompted the creation of “softwares” that allows to merge the areas observed in the MRI with the transrectal ultrasound (TRUS) images observed in real time [10]. Image fusion prostate biopsy is an effective tool that allows for increased detection of csPCa and decreased detection of iPCa, regardless of the transrectal or transperineal route used [11]. These fusion software platforms facilitate guided biopsies of the suspicious areas previously identified by mpRMN, called targeted biopsies, and perform systematic biopsies for the rest of the gland.

Differences between devices mainly concern the registration algorithm (rigid versus elastic), the navigation strategy (organ-based versus electromagnetic tracking), post-biopsy needle position documentation and the use of articulating robotic arms. This fusion process is performed by computerized image registration, which consists of an overlay of MRI-detected lesions and real-time TRUS images [12]. In this procedure, separate MRI and ultrasound images for prostate biopsy are spatially aligned using a fusion software. There are several commercially available software-assisted MRI–TRUS fusion platforms that use different recording methods (rigid or elastic) to fuse MRI and ultrasound. Rigid registration allows the surgeon to manually rotate the MRI and TRUS images relative to each other to produce the best alignment between the images, even though the images themselves do not change. In contrast, elastic registration uses a software algorithm to compensate for the changes in the shape of the segmented prostate gland, which can occur between preoperative MRI and intraoperative imaging during prostate biopsy [13].

The 2019 European Association of Urology (EAU) PCa guidelines recommend transperineal biopsy as the first option over transrectal biopsies. This recommendation is based on a meta-analysis of seven studies that have included 1330 biopsied men with suspected PCa and have shown that transperineal biopsies significantly reduce complications from infections compared to transrectal biopsies [14]. Given this new context, it is essential to recognize the suitability of the different “software” used for transperineal biopsy via image fusion currently available, as well as to generate new evidence on the complementarity of targeted and systematic biopsies, which has been established using the transrectal route in the systematic review of the Cochrane organization [15].

We perform a systematic review of the literature and Google research with the aim of identifying the currently available commercialized software image-fusion systems for performing prostate biopsies via the transperineal route, determining their effectiveness in the diagnosis of csPCa, and revealing the complementarity of the targeted and systematic transperineal biopsies.

## 2. Evidence Acquisition

### 2.1. Systematic Review

A literature search was carried out by two independent authors (NEP and JM) on the PubMed, Cochrane and Google Scholar databases for articles published before 28 February 2023. The MeSH (Medical Subject Heading) [16] terms ‘prostate cancer’ AND ‘biopsy’ AND ‘MRI’ as well as the keywords ‘prostate cancer’ AND ‘softwares’ OR ‘robot’ OR ‘elastic fusion systems’ OR ‘rigid fusion systems’ AND ‘transperineal’ OR ‘multiparametric MRI fusion biopsy’ AND ‘systematic prostate biopsy’ AND ‘targeted prostate biopsy’ were used. The PRISMA (Preferred Reporting Items for Systematic Reviews and Meta-Analyses) [17] were followed. The PICO (Population, Intervention, Comparison, and Outcomes) [18] selection criteria were established as men who underwent prostate biopsy through the transperineal route using software fusion of ultrasound and multiparametric MRI evaluating the results in the detection of csPCa. Studies not published in English, studies that did not report results, systematic reviews, and meta-analyses were excluded. Discrepancies between the two reviewers were resolved by discussion or by consulting a third reviewer. This review was registered in the international prospective registry of systematic reviews (PROSPERO: CRD42022375619).

A total of 321 references were initially identified, however, 83 of them were first excluded due to duplicated titles. After the analysis of abstracts, another 50 were discarded because they were narrative or systematic reviews and meta-analyses, and then 145 studies, analyzed via the transrectal approach or cognitive transperineal technic, were removed as well. Finally, 43 articles were identified, which complied reports on series of biopsies performed using the image fusion software through the transperineal route; 30 out of these 43 met the PICO selection criteria. The PRISMA study selection flowchart is summarized in Figure 1.

### 2.2. Quality Assessment of Selected Studies

The quality of the selected studies and the risk of bias and applicability concern of single studies included were assessed using the tool, Quality Assessment of Diagnostic Accuracy Studies (QUADAS-2) [19], as shown in Table 1 and Figure 2. We found the heterogeneity of the populations analyzed in the different types of studies selected as a notable reason for the bias of this review. Of the 30 articles, 13 collected data according to the Standards of Reporting for Magnetic Resonance Directed Biopsy (START) Studies of the Prostate working group guidelines [20].

### 2.3. Data Extraction in the Selected Studies

The selected articles were recorded on a standard form, along with information on participants, MRI, and biopsy characteristics. The form included the following details. (1) Article: origin (authors, and year of publication), journal, study design, number of participants, and definition of csPCa. (2) Study participants: pre-biopsy population (biopsy naïve and/or prior negative biopsy and/or AS an/or FT control and/or SBRT recurrence), age, mean prostate volume and mean PSA, PSA density and DRE. (3) MRI: type of magnetic resonance, Tesla, type of coil (surface vs. endorectal), time between MRI and biopsy and PI-RADS version. (4) Biopsy: type of anesthesia, approach, patient position, platform used, fusion mode, probe manipulation, prostate segmentation, PCa and csPCa detection rates, number of cores and number of lesions, PI-RADS score, positive cores in TB and SB and added value SB. The study time and complications were also analyzed.

### 2.4. Searching Devices for MRI–TRUS Fusion Prostate Biopsies in Google and Their Characteristics

In addition to the literature review, a Google^®^ search was conducted to identify all robots and software available to perform transperineal fusion prostate biopsies. We found 11 different types of robots. Table 2 describes their characteristics.

## 3. Evidence Synthesis

The full texts of the 30 selected articles were analyzed, and their results were structured using demographic data, biopsy and MRI characteristics, definition of csPCa, cancer detection rates, and software used.

Series from all the platforms mentioned in Table 2 have been found, except for the articles about prostate biopsies via transperineal fusion performed with Robot Artemis. It is necessary to clarify that there are currently other intelligent ultrasounds with the capacity to do fusion images with resonance (not only for prostate biopsies), but either the evidence was demonstrated for biopsies by the transrectal route, or no evidence was found for the transperineal route in this review.

Of the 30 articles, 21 studies were conducted prospectively (none randomized) [21,22,24,27,28,29,30,32,33,34,35,38,39,40,41,42,43,48,49,50] and nine retrospectively [23,25,26,31,36,37,44,45,47].

Eight series were multicenter [21,24,25,26,34,35,40,49] while the remaining 22 were from a single center [22,23,27,28,29,30,31,32,33,36,37,38,39,40,42,43,44,45,46,47,48,50]. Overall, the characteristics and biopsy results of 11,313 patients were analyzed.

### 3.1. Demographics

The pooled mean age was 66.5 years (63–69). The mean pooled PSA level was 7.8 ng/mL (5.7–10.8). The combined mean prostate volume was 45.4 cc. (34–56). The mean pooled PSA density was 0.17 (0.12–0.27). A total of 10,457 lesions were analyzed. PI-RADS 1–2: 764 lesions, PI-RADS 3: 2748 lesions, PI-RADS 4: 4126 lesions, and PI-RADS 5: 2339 lesions. Likert 1–2: 12 lesions, Likert 3: 111 lesions, Likert 4: 195 lesions and Likert 5: 162 lesions. Information regarding the result of the DRE has been reported by 10 articles [24,30,31,32,36,37,40,41,43,50] with an average abnormal DRE of 22% (5–35).

The study population included two articles [42,45] on patients with naïve biopsy, three articles [34,35,48] on patients with previous negative biopsy, twelve articles [22,23,24,26,28,30,32,33,36,41,43,44] on patients with naïve biopsy and previous negative biopsy, eleven studies [27,29,31,37,38,39,40,46,47,49,50] on patients with naïve biopsy and previous negative biopsy and patients in active surveillance (AS) and finally, two articles by Jacewicz et al. [21,25] on patients with no prior biopsy, patients with prior negative biopsy, AS, focal therapy (FT) control, and SBRT recurrence. These data are presented in Table 3.

### 3.2. Characteristics of the Biopsies and MRI

All resonances performed prior to the biopsies were carried out via 1.5 or 3.0 Tesla resonators scans and interpreted using the PI-RADS score Version 1.0 in 1 article [34], version 2.0 in 26 studies [21,22,23,24,25,27,28,29,30,31,32,33,35,36,37,38,39,40,41,42,43,44,46,47,48,50], and version 2.1 in 1 article [45]. In two series [26,49], the reading of the resonance was carried out using the Likert scale.

All biopsies were performed in the lithotomy position. The biopsies were performed using local anesthesia in 11 series [21,24,25,27,37,38,39,46,47,49,50], 15 with general anesthesia [22,23,26,28,30,31,32,33,34,35,36,40,41,43,44], 1 with sedation [48], and 2 with spinal anesthesia [29,42]. One series did not report the type of anesthesia used [45].

Segmentation of the prostate is not a commonly reported step in fusion biopsy series. In our review, 16 series reported who previously segmented the prostate before performing the image fusion. In 11 studies [24,27,32,38,40,42,43,44,46,47,49], the segmentation was carried out by the urologist, in 3 it was carried out by the radiologists [22,23,33], while in the series by Jacewicz et al. [21,25] the segmentation was carried out by both faculties.

The distribution of the types of robots used in each study can be seen in Table 4. Of the 30 articles analyzed, the biopsies have been carried out using software that performs elastic fusion in 16 studies [21,22,23,25,27,30,32,37,39,40,42,44,45,46,47,50] and in 14, by rigid fusion [24,26,28,29,30,31,33,34,35,36,38,41,43,48,49]. Görtz et al. [30] made a comparison of both fusion systems using the Biopsee (rigid) and the Uronav (elastic) systems. The number of cores for target biopsy ranged from 2 to 8 and from 12 to 24 for systematic biopsy. The biopsy time, specifically called the probe time by the authors, was reported in only five articles [21,25,42,46,49], and was approximately between 15 and 30 min for each biopsy. The total time between anesthesia, positioning, segmentation, and probe time was mentioned by Jacewicz et al. [21,25] and Kozel et al. [46] and they both coincided in about 60 min; see Table 5.

### 3.3. Significant Clinically Prostate Cancer Definition

CsPCa was defined in different ways by the authors: (i) ISUP-GG ≥ 2 [21,22,23,25,27,28,30,31,32,33,34,36,37,39,40,41,42,43,44,45,46,47,48,49,50], (ii) ISUP-GG ≥ 2 or ≥3 positive cores with ≥50% of extension [24], (iii) using the Epstein criteria [29], (iv) ISUP-GG ≥ 2 or Gleason 6 with MCCL (Maximum cancer core length) ≥ 4 mm [35,38], (v) Gleason ≥ 4 + 3 or any grade with MCCL ≥ 6 mm, and (vi) Gleason ≥ 3 + 4 or any ISUP-GG with MCCL ≥ 4 mm [26]. Table 6 presents the definition details.

### 3.4. Cancer Detection Rates (CDR)

Although the sample is heterogeneous due to the mentioned characteristics, the overall CDR for all prostate cancers was 61.4%, while for csPCa it was 47.8%.

The CDR of csPCa according to the PI-RADS category of lesions was 24.5% for PI-RADS 3 lesions, 55.9% for PI-RADS 4, and 81.5% for PI-RADS 5.

The added value of systematic biopsy, defined as the diagnosis of csPCa only in systematic biopsy, was reported in only 14 articles [21,23,24,26,27,29,30,32,34,37,40,45,46,47], with a mean of 9.5%.

### 3.5. Cancer Detection according to Population and Software

To standardize our review, we grouped the prostate cancer detection results according to the population studied and the software used. Table 7 (elastic) and Table 8 (rigid) present the results.

When we analyze our series according to the type of software used to perform the biopsy, we observe that with elastic fusion robots a CDR of 64.8% is obtained for all prostate cancers and for csPCa, it is 50.3%. In contrast, with rigid fusion robots, the CDR for PCa was 57.9% and for csPCa, it was 45.4%. Comparing the same populations between elastic and rigid software, we found that in the biopsy-naïve and prior negative biopsy population, the detection rate of global prostate cancer and csPCa with the elastic fusion biopsy was 53% and 41.4%, respectively, and with the rigid biopsy it was 57.8% and 43.6% respectively. On the other hand, when comparing the biopsy-naïve and prior negative biopsy and the AS populations, the global CDRs and csPCa in the elastic biopsies were 73.2% and 54.9%, respectively, and in the rigid biopsies, they were 69.7% and 61.1%, respectively.

## 4. Discussion

In 2009, Ahmed et al. argued that patients undergoing a TRUS guided biopsy were at risk for a sub-diagnosis of clinical csPCa and an over-diagnosis of iPCa [51]. In 2017, the PROMIS trial finally provided level 1 evidence to support the use of mpMRI before prostate biopsy. This study showed that mpMRI had a reported sensitivity, specificity, positive predictive value, and negative predictive value of 93% (95% CI 88–96), 41% (95% CI 36–46), 51% (95% CI 46–56) and 89% (95% CI 83–94), respectively. The comparison of this data showed that the sensitivity of mpMRI was significantly better than transrectal systematic biopsy: 93% (95% CI 88–96%) versus 48% (95% IC 42–55). The PROMIS trial also showed that mpMRI could avoid 27% of unnecessary biopsies [52]. Following the PROMIS trial results, the PRECISION trial was conducted, which evaluated the role of MRI-guided prostate biopsy in improving the detection of clinically significant prostate cancer. This study showed a higher rate of detection of csPCa in the targeted biopsy group over the systematic biopsy group (38% vs. 26%, respectively) [53].

Despite the findings of the PRECISION trial, the additional diagnostic yield of retaining routine systematic prostate biopsy performed in conjunction with mpMRI-targeted strategies remains a matter of debate [54]. Multicenter studies, such as MRI-FIRST trial, 4 M trial and PAIREDCAP trial, in which the role of systematic 12-core transrectal biopsy and the MRI-targeted biopsies was evaluated, showed similar results in that the greatest yield from biopsy is obtained when both types of biopsies are performed [54,55,56,57].

Petov et al. have recently published a meta-analysis which concluded that robotic biopsy on targeted and systematic biopsy have comparable csPCa and overall detection rates [58]. In line with the results of these studies, the Cochrane systematic review makes clear its position regarding the complementarity of systematic biopsies with targeted biopsies via the transrectal route [15]. However, the complementarity of both types of biopsies via the transperineal route is not yet established. Recently, Porpiglia et al. [59] compared the detection rate of csPCa between targeted biopsy alone and targeted biopsy complemented with systematic biopsy. The fusion biopsy was performed with an unusual scheme in naïve biopsy men with positive mpMRI. If the PIRADS 4–5 lesion was anterior, a transperineal approach was used. If the PIRADS 4–5 lesion was posterior, a transrectal approach was used. This trial was designed as a non-inferiority study and concluded that fusion-targeted biopsy alone was not inferior than the fusion biopsy complemented with systematic biopsy for the detection of csPCa.

The value of targeted fusion-biopsy is widely acknowledged as MRI–TRUS fusion-biopsies detect more csPCa than the conventional 12-core TRUS systematic biopsies [51]. Among patients with MRI-visible prostate lesions, the addition of MRI-targeted biopsy to systematic biopsy increased the detection of csPCa and led to a net decrease in the detection of iPCa. Although many of these benefits resulted from MRI-targeted biopsy alone, omission of systematic biopsy would have led to missing the diagnosis of 8.8% of csPCa [52]. In our review, the detection of csPCa only in systematic biopsy was reported in 14 articles, which is an average of 9.5%, ranging from 0.8% to 22.1%. This significant difference may be due to multiple factors typical of the correct characterization of mpMRI, but also due to the inexperience of the surgeon, generating imprecision when establishing suspicious lesions [60].

Regarding the biopsy approach route, since 2019, the European Association of Urology (EAU) PCa guidelines recommend transperineal biopsy as the first option over transrectal biopsies. Tewes et al. compared fusion-guided transrectal biopsies with the transperineal route and found that the rate of targeted biopsy PCa detection rate was 39% via the transrectal route and 75% via the transperineal route [61]. Pepe et al. reported that the transperineal cognitive targeted biopsy detected a significantly higher percentage of csPCa from the anterior area compared to the transrectal directed biopsy [62]. The recent publication by Zattoni et al., reporting a multicenter study of 5241 fusion biopsies from both access routes, concluded that target biopsies via the transperineal route improve the detection of csPCa compared to targeted biopsies via the transrectal route, especially in the apex, transition/central zone, and anterior zones [63].

In this context of precision medicine, for the most sophisticated and accurate detection of csPCa, we have carried out, to our knowledge, the first systematic review that analyzes the diagnostic effectiveness of the currently available software and robots for MRI–TRUS fusion targeted biopsies through the transperineal route. This review showed that prostate biopsies performed guided by the image fusion robots through the transperineal route exhibits a higher detection rate of csPCa compared with the traditional systematic transrectal prostate biopsy. All the devices analyzed in this systematic review are approved by the US Food and Drug Administration (FDA) and the European Medicines Agency (EMA). Of all the robots reported in this article, Artemis (Eigen, Inc.) is the only one that, to date, has no articles evaluating its results via the transperineal approach. In this review, we found no differences between the robots used and the PCa detection rate. Differences between devices mainly concern the registration algorithm (rigid vs. elastic), navigation strategy (organ-based versus electromagnetic tracking), post biopsy needle position documentation and use of articulating robotic arms. Venderink et al. [64] did not identify a significant difference between rigid and elastic images for MRI–TRUS fusion-guided biopsy in csPCa detection. However, the review was carried out through series in which, for the most part, the approach was transrectal. Gortz et al. [30] carried out the first comparative and randomized study between two series of transperineal fusion biopsy performed with rigid biopsy software and another with elastic biopsy. Their results concluded that transperineal MRI/TRUS targeted biopsies directed with a rigid image registration system showed a significantly higher PCa detection rate than elastic TB. We also observed this trend in our analysis of the data, which was sorted based on the types of populations biopsied and the type of biopsy used (rigid vs. elastic). Our findings revealed that transperineal rigid biopsy led to a higher detection rate of clinically significant prostate cancer.

In the latest studies that have been reviewed in this article, there is evidence of a tendency in most recent publications on prostate biopsies via image fusion to be not only mainly transperineal but also to be conducted under local anesthesia, demonstrating the feasibility of this technique with very good results. In addition, novel transperineal biopsy techniques are being developed with existing software platforms to further improve detection rates of csPca using local anesthesia via the transperineal route. Such is the case of the recent publications by Fletcher et al. [49] with Vector Biopsy and by Kaneco et al. [50] with the Double-Freehand Technique.

This review has some limitations. The populations analyzed are very heterogeneous. To somewhat lessen the impact of this breadth of populations analyzed, a homogenization was performed based on the population characteristics and through the rigid vs. elastic software used. Additionally, the series analyzed are not free of bias. The collection of the sample is not uniform, and the criteria for the interpretation of clinically significant prostate cancer vary from one series to another. Not all series used the START criteria for their structure. The experience of surgeons when performing biopsies was reported in a very few series and has a notable impact on the interpretation of the images to be segmented, on the manipulation of the robot, and on the degree of cognition for interpretation. Moreover, the segmentation of the prostate is a step rarely reported in the series and we believe that its correct performance is fundamental for the correct result of the biopsy.

## 5. Conclusions

This systematic review shows that the software analyzed for performing prostate biopsies directed via transperineal image fusion constitutes an effective method for the detection of csPCa. Targeted transperineal MRI–TRUS fusion prostate biopsies using a rigid image registration system showed a higher csPCa detection rate than those using elastic image registration. No diagnostic differences between the different types of robotic systems currently available were observed. However, one spread robotic system has not been analyzed. The complementarity of systematic biopsy has also been demonstrated in the transperineal MRI–TRUS image fusion biopsies, regardless of the fusion prototype used.

## Figures and Tables

**Figure 1 cancers-15-03329-f001:**
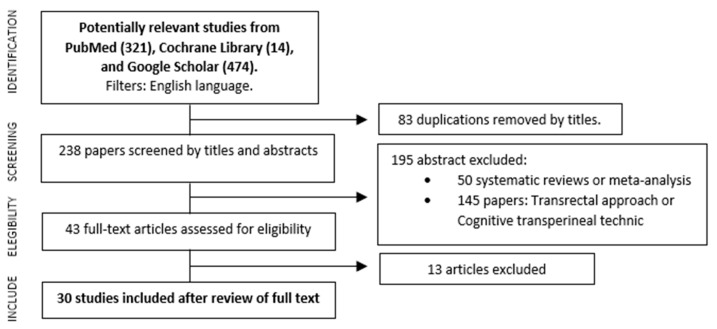
Flowchart of the search strategy.

**Figure 2 cancers-15-03329-f002:**
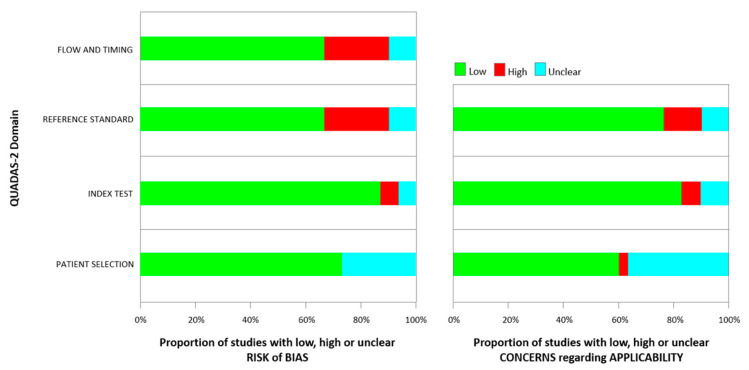
Quality assessment according to QUADAS-2 of the included studies.

**Table 1 cancers-15-03329-t001:** Risk of bias assessed using QUADAS−2.

Study	Risk of Bias				Applicability Concerns		
	Patient Selection	Index Test	Reference Standard	Flow and Timing	Patient Selection	Index Test	Reference Standard
Jacewlcz et al. [21]	☺	☺	☹	☹	☺	☺	☺
Mischinger et al. [22]	☺	☺	☺	☺	☺	☺	?
Lee et al. [23]	?	☺	?	☺	?	☺	☺
Marra et al. [24]	☺	☺	☺	☺	☺	?	☺
Jacewicz et al. [25]	☺	☺	☹	☹	☺	☺	☺
Miah et al. [26]	?	☹	☺	☺	?	☺	☺
Günzel et al. [27]	☺	☺	☺	☺	☺	☹	☺
Mehmood et al. [28]	☺	☹	☹	☺	☹	☺	☺
Hakozaki et al. [29]	☺	☺	☺	☺	☺	☺	☹
Görtz et al. [30]	☺	☺	☺	☹	☺	☺	☺
Hakozaki et al. [31]	?	☺	☹	☺	?	☺	☺
Miah et al. [32]	☺	?	☺	☺	☺	?	☺
Tschirdewahn et al. [33]	☺	☺	☺	☹	☺	☺	☺
Hansen et al. [34]	☺	☺	☺	☹	☺	☺	?
Lian et al. [35]	☺	☺	☹	☺	☺	☺	☺
Radtkeet al. [36]	?	☺	☺	☺	?	☺	☺
Kim et al. [37]	?	☺	☺	☺	?	☺	L
De Vulder et al. [38]	☺	☺	☺	☺	☺	☺	☺
Winoker et al. [39]	☺	?	☺	?	☺	?	☺
Wajswol et al. [40]	☺	☺	☺	?	?	☺	?
Hansen et al. [41]	?	☺	☹	☺	☺	☺	☺
Shoji et al. [42]	☺	☺	☺	☹	☺	☺	☺
Kaufmann et al. [43]	☺	☺	☺	☹	☺	☺	☹
Fulco et al. [44]	☺	☺	☹	☺	?	☺	☺
Thaiss et al. [45]	?	☺	☺	☺	?	☺	☺
Kozel et al. [46]	☺	☺	?	☺	☺	☺	☺
Dahl et al. [47]	?	☺	?	☺	?	☺	☺
Pepe et al. [48]	☺	☺	☺	☺	☺	☹	☹
Fletcher et al. [49]	☺	☺	☺	☺	?	☺	☺
Kaneko et al. [50]	☺	☺	☺	?	?	☺	☺

☺ Low Risk; ☹ High Risk; ? Unclear Risk.

**Table 2 cancers-15-03329-t002:** Types of robots currently available. Google^®^ search.

Fusion Biopsy System (Manufacturer)	Ultrasound Image Acquisition	Ultrasound Tracking Mechanism	Fusion Method	Biopsy Route
Artemis (Eigen, Inc., Grass Valley, CA, USA)	Manual rotation along a fixed axis (ultrasound probe on a tracking arm).	Mechanical arm with encoded joints.	Rigid/Elastic	TP/TR
BioJet (D&K Technologies, Inc., Barum, Germany)	Real-time biplanar TRUS and 3D model of the prostate mounted on a positioning system.	Stepper with 2 built-in encoders.	Rigid/Elastic	TP/TR
Biopsee (MedCom, Inc., Darmstadt, Germany )	Custom-made biplane TRUS probe mounted on a stepper.	Stepper with 2 built-in encoders.	Rigid/Elastic	TP/TR
HI RVS/Real-time Virtual Sonography (Hitachi, Inc., Tokio, Japan)	Real-time biplanar TRUS.	Electromagnetic tracking.	Rigid	TP/TR
UroNav (In Vivo/Philips, Inc., Cambridge, MA, USA)	Manual ultrasound 2D sweep. Freehand manipulation of ultrasound probe or mounted on a stepper.	Electromagnetic tracking ultrasound.	Rigid/Elastic	TP/TR
Urostation (Koelis, Inc., Meylan, France)	Automatic ultrasound probe rotation, three different volumes elastically registered.	Image-based registration.	Elastic	TP/TR
iSR’obot Mona Lisa (Biobot Surgical, Inc., Singapore)	Motorized translation.	Robotic arm.	Elastic	TP
MIM Symphonyx Bx (BK, Inc., Seoul, South Korea)	Motorized translation.	Encoder.	Rigid	TP/TR
Virtual Navigator (Esaote, Inc., Genova, Italy)	Manual ultrasound sweep. Freehand rotation of ultrasound probe.	Electromagnetic tracking ultrasound and needle.	Rigid	TP/TR
Aplio i800 (Canon Medical Systems, Inc., Crawley, UK)	Freehand sweep.	Fusion Imaging of live US/MRI guidance for targeted biopsy.	Rigid	TP/TR
Logiq (GE, Inc., Boston, MA, USA)	Freehand sweep.	Fusion Imaging of live US/MRI guidance for targeted biopsy.	Rigid	TP/TR

**Table 3 cancers-15-03329-t003:** Characteristics of the studies and populations included.

Serie	Author	Country	Study Design	N Centers	N Patients	Pop.	Mean Age (yr)	Mean PSA (ng/mL)	Mean PV (cc)	Mean PSAd	DRE+ (%)
1	Jacewlcz et al., 2021 [21]	Norway	Prospective	2	401	4	69	6.9	40	0.17	NR
2	Mischinger et al., 2018 [22]	Germany	Prospective	1	202	2	66	8	36	0.21	NR
3	Lee et al., 2020 [23]	Singapore	Retrospective	1	433	2	66.1	10.4	43.2	0.27	NR
4	Marra et al., 2021 [24]	Italy	Prospective	4	1014	2	66.8	8.1	51.3	0.2	23.4
5	Jacewicz et al., 2020 [25]	Norway	Retrospective	2	377	4	67	7.2	43	0.17	NR
6	Miah et al., 2019 [26]	UK	Retrospective	11	640	2	63.8	7.8	47.4	0.16	NR
7	Günzel et al., 2022 [27]	Norway	Prospective	1	969	3	68	6.72	45	0.15	NR
8	Mehmood et al., 2021 [28]	KSA	Prospective	1	100	2	64	6.1	50	0.12	NR
9	Hakozaki et al., 2017 [29]	Japan	Prospective	1	177	3	68	7.42	42.9	0.17	28.8
10	Görtz et al., 2022 [30]	Germany	Prospective	1	939	2	65	7.7/7.6 *^1^	44/50 *^1^	0.17/0.15 *^1^	35/28 *^1^
11	Hakozaki et al., 2019 [31]	Japan	Retrospective	1	310	3	68.2	8.6	42.8	0.12	20.9
12	Miah et al., 2019 [32]	UK	Prospective	1	86	2	64.2	10	51.03	0.19	NR
13	Tschirdewahn et al., 2020 [33]	Germany	Prospective	1	213	2	66	7.8	50	0.14	15
14	Hansen et al., 2017 [34]	Germany	Prospective	2	487	5	66	9	56	0.15	NR
15	Lian et al., 2017 [35]	China	Prospective	2	101	5	68.9	10.8	42.1	0.25	NR
16	Radtke et al., 2015 [36]	Germany	Retrospective	1	191	2	66	7.9	44	0.19	20.9
17	Kim et al., 2022 [37]	USA	Retrospective	1	301	3	67	6	45	0.14	5
18	De Vulder et al., 2022 [38]	Belgium	Prospective	1	203	3	69	6.8	49	0.16	NR
19	Winoker et al., 2020 [39]	USA	Prospective	1	168	3	68	7.9	49	0.16	NR
20	Wajswol et al., 2020 [40]	USA	Prospective	1	176	3	67.5	8.25	41	0.2	27.8
21	Hansen et al., 2017 [41]	UK	Prospective	3	807	2	65	6.5	42	0.15	23
22	Shoji et al., 2017 [42]	Japan	Prospective	1	250	1	68	6.7	34	0.19	NR
23	Kaufmann et al., 2021 [43]	Switzerland	Prospective	1	392	2	64	7	43	0.16	16
24	Fulco et al., 2020 [44]	Italy	Retrospective	1	272	2	68	7.2	NR	NR	NR
25	Thaiss et al., 2021 [45]	Germany	Retrospective	1	563	1	66	9.8	45	0.21	NR
26	Kozel et al., 2022 [46]	USA	Prospective	1	200	3	67	7.89	52	0.15	NR
27	Dahl et al., 2022 [47]	USA	Retrospective	1	301	3	66	5.7	45	0.15	NR
28	Pepe et al., 2019 [48]	Italy	Prospective	1	875	5	63	9.8	44.6	0.21	NR
29	Fletcher et al., 2023 [49]	UK	Prospective	2	69	3	67	7.9	43	0.16	NR
30	Kaneko et al., 2023 [50]	USA	Prospective	1	96	3	68	7.84	56	0.13	21

Pop: Population. PV: Prostate Volume. PSAd: Density of PSA (PV/PSA). DRE+: Pathologic Digital Rectal Exam. NR: Not Reported. *^1^: compared the results between two platforms: Biopsee/Uronav. Population 1: Biopsy Naïve. Population 2: Biopsy Naïve + Prior negative biopsy. Population 3: Biopsy Naïve + Prior negative biopsy + AS. Population 4: Biopsy Naïve + Prior negative biopsy + AS+ FT control + SBRT recurrence. Population 5: Prior negative biopsy.

**Table 4 cancers-15-03329-t004:** Robots and platforms used.

Serie	MRI Tesla	MRI type	PIRADS Version	Coil Type	Anesthesia	Approach	Patient Position	Platform Used	Fusion Mode	Probe Manipulation	Prostate Segmentation
1	1.5	Biparametric	v2.0	Surface	Local	TP	Lithotomy	Koelis	Elastic	Freehand	Both
2	1.5 or 3.0	Multiparametric	v2.0	Endorectal	General	TP	Lithotomy	iSR’obot Mona Lisa Urofusion	Elastic	Robotic arm	Radiologist
3	3.0	Multiparametric	V2.0	Endorectal	General	TP	Lithotomy	iSR’obot Mona Lisa Urofusion	Elastic	Robotic arm	Radiologist
4	1.5 or 3.0	Both	v2.0	Endorectal	Local	TP	Lithotomy	Esaote	Rigid	Freehand	Urologist
5	1.5 or 3.0	Both	v2.0	Surface	Local	TP	Lithotomy	Koelis	Elastic	Steady pro arm	Both
6	1.5 or 3.0	Multiparametric	Likert	NR	General	TP	Lithotomy	MIM-Symphony-DX	Rigid	Encoders with brachytherapy template	NR
7	1.5 or 3.0	Multiparametric	v2.0	Endorectal	Local	TP	Lithotomy	Koelis	Elastic	Steady Pro arm	Urologist
8	3.0	Multiparametric	v2.0	NR	General	TP	Lithotomy	Biojet	Rigid	Robotic arm	NR
9	3.0	Multiparametric	v2.0	Surface	Spinal	TP	Lithotomy	(RVS) system Hitachi	Rigid	Electromagnetic tracking/Freehand	NR
10	3.0	Multiparametric	v2.0	Surface	General	TP	Lithotomy	Biopsee/Uronav	Rigid/ Elastic	Freehand	NR
11	3.0	Multiparametric	v2.0	Surface	General/Spinal	TP	Lithotomy	(RVS) system Hitachi	Rigid	Electromagnetic tracking	NR
12	3.0	Multiparametric	v2.0	Surface	General	TP	Lithotomy	iSR’obot Mona Lisa Urofusion	Elastic	Robotic arm	Urologist
13	3.0	Multiparametric	v2.0	Endorectal	General	TP	Lithotomy	MIM-Symphony-DX	Rigid	Encoders with brachytherapy template	Radiologist
14	1.5 or 3.0	Multiparametric	v1.0	Endorectal	General	TP	Lithotomy	Biopsee	Rigid	Freehand	NR
15	3.0	Multiparametric	v2.0	Endorectal	General	TP	Lithotomy	(RVS) System Hitachi	Rigid	Electromagnetic tracking/Freehand	NR
16	3.0	Multiparametric	v2.0	Endorectal	General	TP	Lithotomy	Biopsee	Rigid	Freehand	NR
17	3.0	Multiparametric	v2.0	Surface	Local	TP	Lithotomy	Uronav	Elastic	Electromagnetic tracking/Freehand	NR
18	3.0	Multiparametric	v2.0	Surface	Local	TP	Lithotomy	Aplio i800	Rigid	Freehand	Urologist
19	1.5 or 3.0	Multiparametric	v2.0	Surface	Local	TP	Lithotomy	Uronav	Elastic	Electromagnetic tracking/Freehand	NR
20	3.0	Multiparametric	v2.0	Surface	General	TP	Lithotomy	Uronav	Elastic	Electromagnetic tracking/Freehand	Urologist
21	3.0	Multiparametric	v2.0	NR	General	TP	Lithotomy	Biopsee	Rigid	Encoders with brachytherapy template	NR
22	3.0	Multiparametric	v2.0	NR	Spinal	TP	Lithotomy	Biojet	Elastic	Robotic arm	Urologist
23	3.0	Multiparametric	v2.0	Surface	General	TP	Lithotomy	Biopsee	Rigid	Encoders with brachytherapy template	Urologist
24	1.5	Multiparametric	v2.0	Endorectal	Local/General	TP	Lithotomy	Koelis	Elastic	Steady pro arm	Urologist
25	3.0	Both	v2.1	Surface	NR	TP	Lithotomy	iSR’obot Mona Lisa Urofusion	Elastic	Robotic arm	NR
26	3.0	Multiparametric	v2.0	NR	Local	TP	Lithotomy	Uronav	Elastic	Electromagnetic tracking/Freehand	Urologist
27	3.0	Multiparametric	v2.0	Surface	Local	TP	Lithotomy	Uronav	Elastic	Electromagnetic tracking/Freehand	Urologist
28	3.0	Multiparametric	v2.0	NR	Sedation	TP	Lithotomy	GE logic	Rigid	Electromagnetic tracking/Freehand	NR
29	1.5 or 3.0	Multiparametric	Likert	Surface	Local	TP	Lithotomy	Vector Biopsy: Biopsee + VirtuTRAX EM	Rigid	Electromagnetic tracking	Urologist
30	3.0	Multiparametric	v2.0	NR	Local	TP	Lithotomy	Koelis 3D Models	Elastic	The Double-Freehand Technique	NR

Endorectal Coil: whether endorectal coil was used or not when the MRI was performed. NR: Not Reported. Prostate Segmentation: who was the specialist who performed the segmentation and planning of the prostate prior to the biopsy.

**Table 5 cancers-15-03329-t005:** Characteristics of the biopsies.

Number	N° PIRADS 1–2	N° PIRADS 3	N° PIRADS 4	N° PIRADS 5	N° Cores p/Lesion	N° Cores Systematic	N° Total Cores	Mean Lesion Volume (mL)	Mean MCCL (mm)
1	53	74	137	137	4–5	10–12	NR	NR	8
2	39	39	107	17	5.8 (mean)	NR	NR	NR	NR
3	NR	71	254	108	5	24.5 (mean)	34	NR	NR
4	NR	599	657	144	2–4	10–12	15.3 (mean)	0.5	NR
5	22	97	142	115	1–4	6–10	NR	0.9	9
6	NR	98 (L) *^1^	173 (L) *^1^	140 (L) *^1^	4–6	16.3	NR	NR	7
7	24	45	206	438	4–6	10–12	NR	NR	NR
8	20	17	27	36	2–4	12	NR	NR	NR
9	NR	NR	NR	NR	4–6	12	18	NR	NR
10	NR	367/310 *^2^	333/332 *^2^	139/124 *^2^	4/5 *^2^	24/25 *^2^	29/34 *^2^	NR	NR
11	NR	NR	NR	NR	4	15	20	NR	NR
12	NR	22	55	30	8.1 (mean)	20.2 (mean)	NR	NR	8
13	NR	99	78	25	9	24	33	NR	NR
14	144	128	100	115	3	24	27	NR	NR
15	13	31	36	21	4.2	12	16–18	NR	NR
16	13	50	81	50	3	24	27	NR	NR
17	NR	62	151	88	3–4	20	24	NR	NR
18	NR	56	121	46	5	10	15	NR	NR
19	NR	26	76	66	3–4	12	16–18	NR	NR
20	14	53	128	73	4–5	12	16–18	NR	NR
21	236	153		418 *^3^	4	20	24	NR	NR
22	NR	NR	NR	NR	2–4	12	16–18	NR	NR
23	124	87	123	58	5	35	42	NR	NR
24	NR	115	129	28	3–5	10	15	NR	NR
25	29	95	324	115	4	14	18	3.2	NR
26	6	73	135	51	2–3	12	15	NR	NR
27	NR	62	151	88	3	20	23	NR	NR
28	NR	NR	NR	NR	4	20	30	NR	NR
29	12 (L) *^1^	13 (L) *^1^	22 (L) *^1^	22 (L) *^1^	2–4	12	NR	NR	NR
30	27	17	25	27	2–4	12–14	NR	NR	NR

(L)*^1^: Likert score. *^2^: Gortz et al. [30] compared the results between two platforms: Biopsee/Uronav. *^3^: The authors pooled the results obtained from PIRADS 4 and 5 lesions. NR: Not Reported. MCCL: Maximum cancer core length.

**Table 6 cancers-15-03329-t006:** Results.

Number	csPCa Definition	PCa (%)	csPCa (%)	csPCa (%) SB	csPCa (%) PIRADS 1–2	csPCa (%) PIRADS 3	csPCa (%) PIRADS 4	csPCa (%) PIRADS 5
1	ISUP ≥ 2	65	48	2	17	20	39	77
2	ISUP ≥ 2	61	52	NR	7.6	25.6	72	88
3	ISUP ≥ 2	57	46	17	NR	16	38	85
4	ISUP ≥ 2/≥3 positive cores/≥50% of extension	43.9	39.4	15.9	NR	15.4	46.2	73.9
5	ISUP ≥ 2	64	51	NR	18	20	60	84
6	Ahmed D1. Gleason ≥ 4 + 3 or any grade ≥ 6 mm. Ahmed D2. Gleason ≥ 3 + 4 or any grade ≥ 4 mm.	65	49.8	0.8	NR	21.6 (L) *^1^	47.1 (L) *^1^	84.2 (L) *^1^
7	ISUP ≥ 2	66	49	1.5	9	15	50	73
8	ISUP ≥ 2	45	33.3	NR	NR	NR	NR	NR
9	Epstein criteria	65.5	63.3	14	NR	NR	NR	NR
10	ISUP ≥ 2	62.5/59.5 *^2^	46/46 *^2^	9	NR	NR	NR	NR
11	ISUP ≥ 2	47	43.9	NR	NR	3.1	50.2	80.8
12	ISUP ≥ 2	51.2	40.7	10.5	NR	0	40	53.3
13	ISUP ≥ 2	59	40	NR	NR	NR	NR	NR
14	ISUP ≥ 2	51	39	11	NR	19.5	32	70.4
15	ISUP ≥ 2 or Gleason 6 with MCCL ≥ 4 mm.	40.6	24.8	NR	NR	NR	NR	NR
16	ISUP ≥ 2	72.7	53.9	NR	NR	NR	NR	NR
17	ISUP ≥ 2	79.1	62.2	10.9	NR	33	62	72
18	ISUP ≥ 2 or Gleason 6 with MCCL ≥ 4 mm.	73.5	60.1	NR	NR	23.2	66.1	89.1
19	ISUP ≥ 2	79	59	NR	NR	42.3	76.3	95.5
20	ISUP ≥ 2	76.7	58.5	9.6	28.6	42.3	76.3	95.5
21	ISUP ≥ 2	68	49	NR	NR	30.7		71 *^3^
22	ISUP ≥ 2	58	55	NR	0	6.8	49	80
23	ISUP ≥ 2	51	40	NR	NR	57	83	98
24	ISUP ≥ 2	43	27.2	NR	NR	6.09	34.1	82.1
25	ISUP ≥ 2	59.9	51.1	2.9	NR	6.6	64.7	74.8
26	ISUP ≥ 2	71	58.5	6	NR	NR	NR	NR
27	ISUP ≥ 2	79	49.1	22.1	NR	NR	NR	NR
28	ISUP ≥ 2	34.5	22.5	NR	NR	NR	NR	NR
29	ISUP ≥ 2	92.9	77.1	NR	NR	62	68	95
30	ISUP ≥ 2	62	48	NR	3.7	47		71 *^3^

csPca definition: Definition of clinically significant prostate cancer. PCa (%): Overall prostate cancer detection Rate. csPCa (%): Clinically significant prostate cancer detection rate. csPCa (%) SB: Detection of clinically significant prostate cancer only in systematic biopsy. csPCa (%) PIRADS: Detection of clinically significant cancer according to the PIRADS score. (L)*^1^: Likert score. *^2^: Gortz et al. [30] compared the results between two platforms: Biopsee/Uronav. *^3^: The authors pooled the results obtained from PIRADS 4 and 5 lesions. NR: Not Reported. MCCL: Maximum cancer core length.

**Table 7 cancers-15-03329-t007:** Population and elastic fusion.

N° of Patients	Population	Mean Age (yr)	Mean PSA (ng/mL)	Mean PV (cc)	Mean PSAd	PCa (%)	csPCa (%)
813	Biopsy Naïve	67	8.25	39.5	0.2	58.9	53
993	Biopsy-Naïve + Prior negative biopsy	66	8.9	43.4	0.2	53	41.4
2211	Biopsy-Naïve + Prior negative biopsy + AS	67.3	7.1	47.5	0.15	73.2	54.9
778	Biopsy-Naïve + Prior negative biopsy + AS+ FT control + SBRT recurrence.	68	7	41.5	0.17	64.5	49.5

PV: Prostate volume. PSAd: Density of PSA (PV/PSA).

**Table 8 cancers-15-03329-t008:** Population and rigid fusion.

N° of Patients	Population	Mean Age (yr)	Mean PSA (ng/mL)	Mean PV (cc)	Mean PSAd	PCa (%)	csPCa (%)
1463	Prior negative biopsy	65.9	9.86	47.5	0.2	42	28.7
3357	Biopsy-Naïve + Prior negative biopsy	65	7.3	46.8	0.16	57.8	43.6
759	Biopsy-Naïve + Prior negative biopsy + AS	68	7.6	44.4	0.15	69.7	61.1

PV: Prostate volume. PSAd: Density of PSA (PV/PSA).

## Data Availability

Data are available in PubMed, Cochrane, and Google Scholar databases.

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
