# Peer review of "A Systematic Review of the Current Status of Magnetic Resonance–Ultrasound Images Fusion Software Platforms for Transperineal Prostate Biopsies"

_cancers, 2023, doi:10.3390/cancers15133329_

Round 1

Reviewer 1 Report

The authors wanted to state the reliability of different fusion biopsy-softwares currently available.

I found the paper interesting as it refers to a topic wich has not been treated in the available literature. Moreover the review design confers ad adequate level of evidence.

The paper design is well explained and i found the methjodology correct.

The manuscript is easy-reading and the message the authors wanted to drive is clear to the reader.

I would suggest this manuscript for pubbblication

Author Response

Dear reviewer, thank you very much for your comments and for taking the time to read our work. Kind regards

Reviewer 2 Report

This systematic review is a well organized thesis on the field of prostate biopsy, which has been subject to many studies and controversies. It is thought that it will be of great help to future research for proper diagnosis of prostate cancer, which has recently been one of the most important in the field of prostate cancer.

Author Response

Dear reviewer, thank you very much for your comments and for taking the time to read our work. Kind regards. 

Reviewer 3 Report

I would like to express my appreciation for your well-executed and organized systematic review, which provides accurate standardization criteria and offers valuable insights into the topic. I have just a few comments and suggestions:

  • - I recommend investigating whether the use of digital rectal examination (DRE) has an impact on the detection of clinically significant prostate cancer following biopsy. This investigation could provide valuable insights into the role of DRE in conjunction with multiparametric magnetic resonance imaging (mpMRI).

  • - It would be prudent to exclude studies that utilize the Likert system for evaluation and instead focus solely on those employing the PI-RADS system. This exclusion would help streamline the review and maintain a standardized approach.

  • - While acknowledging the heterogeneity of the populations analyzed in the included studies, it would be valuable to discuss the implications of this heterogeneity on the interpretation of the results. Addressing any potential limitations or confounding factors arising from the diversity of populations would provide a comprehensive perspective on the findings.

  • - It would be interesting to investigate whether there are any differences between fusion biopsy systems that utilize a robotic arm and those performed freehand.

Once again, I commend you on the quality of your review and the valuable insights it provides. I believe these suggestions could further enhance the impact and applicability of your work. Thank you for considering my feedback, and I look forward to seeing the continued advancements in this field.

Author Response

Dear reviewer, thank you very much for your comments and for taking the time to read our work. His suggestions have been very productive for us.

We believe that the use of digital rectal examination (DRE) has been evaluated in few series in our review, so we could not determine the impact on the detection of clinically significant prostate cancer. However, the DRE provides a variety of valuable information: abnormalities on the posterior surface of the prostate gland, which are suggestive of PCa, can be detected, prostate volume can be categorized, and the presence of inflammation can be inferred from the data reported by the patient.

It would be prudent to exclude studies that use the Likert system for assessment and instead focus only on those that use the PI-RADS system. This exclusion would help simplify the review and maintain a standardized approach:
We had considered this previously, but we consider that the studies that have been published with the Likert score are well-conducted studies, with low bias and that they contribute a lot to our review.

While the heterogeneity of the populations analyzed in the included studies is recognized, it would be valuable to discuss the implications of this heterogeneity for the interpretation of the results. Thank you very much for your comment. It is highlighted in red in the text.

- It would be interesting to investigate whether there are differences between fusion biopsy systems that use a robotic arm and those that are performed by freehand: thank you very much, we have found no differences between the series published with the robotic arm and freehand.

Thank you again for your comments and suggestions that have been very constructive for our work,